# Seeing Like Humans: Task-Driven Token Reduction for Accelerated ViT in Robotic Navigation

## Abstract

In robotics, vision is critical for enabling agents to perceive and interact with their environment. Recent advancements in vision models, particularly Vision Transformers (ViTs), have shown remarkable performance in pure vision tasks like object recognition and scene understanding, showing great potential for robotic applications such as object navigation. However, their computational cost grows quadratically with respect to the number of tokens, posing significant challenges for real-time deployment on resource-constrained robotic platforms. To enhance ViT efficiency in robotic tasks, we propose a biologically-inspired token reduction framework that dynamically allocates computation to task-relevant regions in images while neglecting those irrelevant regions for efficiency. Our method introduces two key components: (1) a task-driven spatial attention mechanism that selectively prunes redundant tokens based on the current task, and (2) a temporal feature reusing module that reuses stable visual features across frames to minimize redundant computation. Together, these components enable the visual perception model to focus only on relevant regions, significantly improving inference speed. Experiments show that our method notably reduces inference time in object navigation tasks without significant performance degradation. Additionally, it enables practical ViT deployment on edge devices such as the Jetson Orin (high-performance GPU) and Raspberry Pi 4B (lightweight CPU), achieving 56.5 FPS and 2 FPS, respectively. This represents a $1.5\sim3\times$ speedup over standard ViTs, making real-time robotic vision more feasible.

## 1 Introduction

In robotic systems operating in unstructured environments, real-time visual processing is critical for autonomous decision-making. From mobile robots navigating through crowded spaces to manipulators interacting with deformable objects, the ability to rapidly process visual inputs directly influences both operational safety and task performance. Early robotic vision systems, based on Convolutional Neural Networks (CNNs), were effective at extracting local features from images and were widely adopted for tasks such as manipulation and navigation. However, CNNs struggle to capture long-range spatial relationships and global context, which are necessary for more complex tasks such as understanding object arrangements or large-scale scene layouts.

Vision Transformers (ViTs) address these limitations by using global attention mechanisms to model long-range dependencies. This is especially beneficial in tasks requiring global visual context, such as object rearrangement and navigation, which depend on understanding the overall spatial layout. ViTs provide a more holistic scene understanding, improving decision-making in cluttered and dynamic environments. However, the quadratic complexity of self-attention leads to a trade-off: increasing input resolution improves visual understanding, but also raises inference latency, which in turn reduces control frequency and may affect safety. This challenge highlights the need for more efficient architectures that retain the benefits of ViTs while meeting the real-time requirements of robotics.

Biological vision systems have developed highly efficient methods for processing visual information. For instance, when driving, we focus on reading road signs while ignoring distant trees. Rather than processing the entire visual field equally, our brain selectively prioritizes the most relevant details. Furthermore, our eyes minimize redundant processing by concentrating only on new or changing elements between blinks or saccades, enabling us to efficiently scan our surroundings

without wasting energy. This efficiency allows us to interact with complex environments with minimal energy consumption (Van Essen et al., 1992; Sperling & Melchner, 1978; Thorpe, 1991).

Motivated by these biological principles, we formalize two complementary axioms for efficient visual processing. (1) Temporal Efficiency: Processing of static visual features should be minimized across sequential frames to avoid redundant computation. (2) Spatial Sparsity: Computational resources should focus on regions critical for the current task.

Building on these principles, we propose a token reduction method that mirrors the efficiency of biological vision systems. Our method incorporates two complementary modules: the temporal token reuse module that reduces redundant information across frames, and the task-driven token pruning module that prunes irrelevant regions within each frame. **The temporal token reusing module** identifies stable, persistent features across frames, reducing redundant computation by reusing information from previous time steps. It leverages cross-attention to propagate relevant features, ensuring that only the most dynamic or task-relevant elements are processed. **The task-driven token pruning module**, on the other hand, evaluates the relevance of each token based on the current task and robot state. By focusing on task-relevant regions, such as target objects or navigable paths, it prunes unnecessary tokens, allowing the model to allocate resources more efficiently.

Unlike most prior token reduction methods with fixed pruning ratios and single-goal training, our temporal-reuse and task-driven modules perform token-level pruning guided by temporal redundancy and task-specific requirements, resulting in a dynamic token selection rate that adapts to varying scenes and objectives. This design enables a single model to generalize across multiple goals without retraining, thereby ensuring both efficiency and robustness in diverse environments.

Evaluated on object navigation tasks, our reduction strategy achieves 1.5× speedup on Jetson Orin and nearly 3× speedup on Raspberry Pi 4B compared to standard ViTs. We conducted tests on these platforms because they are commonly used in robotics: the Jetson Orin is a GPU frequently used in high-performance quadrupedal and humanoid robots, while the Raspberry Pi 4B is a popular CPU device in lightweight, cost-effective robots.

In summary, our contributions are as follows:

1. We proposed a task-driven token pruning module that selectively processes relevant tokens while discarding unnecessary features, thereby improving computational efficiency.

2. We introduced a temporal token reuse module that reuses stable visual features across frames, reducing redundant computation and enhancing the model's efficiency during video sequence processing.

3. We demonstrated the effectiveness of our method through extensive experiments, showing significant reductions in inference time and real-time performance across multiple platforms. Specifically, our method achieved 1.5x faster inference on the Jetson Orin and nearly 3x faster performance on the Raspberry Pi 4B.

## 2 RELATED WORK

### 2.1 VISUAL PERCEPTION IN ROBOTICS

Visual perception is crucial for autonomous robots to understand and interact with their environments. Early systems relied on hand-crafted features and later convolutional neural networks (CNNs), which excel at extracting local spatial patterns for tasks such as detection and navigation. However, CNNs struggle to model global context and long-range dependencies, which are essential for complex, unstructured scenarios.

Vision Transformers (ViTs) (Dosovitskiy et al., 2021; Carion et al., 2020) address this limitation by processing images as sequences of patches, enabling global dependency modeling across the entire scene. This capability has benefited various robotics tasks, from grasp detection to navigation and multi-task manipulation (Wang et al., 2022; Majumdar et al., 2023; Brohan et al., 2022; Zitkovich et al., 2023), improving generalization and robustness in novel environments.

Despite these advantages, the quadratic complexity of self-attention leads to high computational cost. Higher image resolutions can enhance perception but also increase inference latency, reducing control

frequency and potentially affecting closed-loop safety. These challenges have driven research into accelerating ViTs for real-time robotic deployment without sacrificing their representational power.

## 2.2 ViT Acceleration Methods

Most acceleration methods for Transformers focus on reducing the computational cost. These methods can be categorized into two main approaches: (1) Reducing the number of tokens. Since the computational complexity of transformers is proportional to the square of the number of tokens, reducing the number of tokens processed can significantly lower the overall computational cost. (2) Modifying the model architecture. These modifications include parameter reduction techniques like pruning and sparsification to eliminate redundant parameters, as well as architectural optimizations of attention mechanisms to reduce computational overhead.

### 2.2.1 Token Pruning Techniques

To determine which tokens to prune, several strategies have been proposed. One approach is prediction-based pruning, where tokens are pruned based on their feature representations at intermediate layers. Tokens that are less informative, as identified by these intermediate features, are discarded early in the pipeline. Methods such as SPViT (Kong et al., 2022) and AdaViT (Meng et al., 2022) apply this technique, using token features to predict which tokens can be safely removed, thus improving efficiency without significantly affecting the final output. Another common approach is attention weight-based pruning, where tokens with lower attention weights are assumed to be less important and are pruned. Fayyaz et al. (2022) and Liang et al. (2022) use attention scores to decide which tokens to discard, effectively reducing the number of tokens processed.

Similarity-based pruning assumes that tokens that are highly similar to each other may contain redundant information and therefore do not all need to be retained. In this approach, tokens that exhibit high similarity in their feature representations are grouped together, with redundant tokens either discarded or merged. This helps reduce the number of tokens while retaining essential information. Techniques like ToME (Bolya et al., 2023) and ALGM (Norouzi et al., 2024) utilize clustering or merging strategies to achieve this goal, improving throughput and efficiency without significantly compromising the model's ability to make accurate predictions.

Once the tokens to be pruned are identified, the next step is determining how to prune them. One straightforward method is to discard the tokens completely. However, this can lead to a loss of important information if the wrong tokens are pruned. Another approach is to use skip connections, where pruned tokens bypass certain layers and are directly passed to the final layers, minimizing unnecessary computations while preserving the important features. For example, DoViT (Liu et al., 2024) applies this method to skip tokens that are flagged as low-confidence. Finally, token merging involves combining similar or redundant tokens into a single token, which helps preserve spatial integrity while reducing the number of tokens. This method has been used in models like ToME (Bolya et al., 2023) and ALGM (Norouzi et al., 2024) to merge tokens at various stages and improve computational efficiency.

*While these pruning techniques are effective for common tasks, their lack of task-awareness often results in a trade-off between speed and performance in dynamic robotic settings.* For instance, if the area of interest is not the main object in the image (e.g., in navigation tasks where subtle obstacles or road conditions are important), these methods may incorrectly prune critical information, leading to a significant drop in task success rate. Similarly, similarity-based pruning often struggles to remove large but irrelevant subjects (e.g., walls or fixed structures), which occupy significant visual space but do not contribute to task completion, failing to reach optimal efficiency. These limitations highlight the need for a pruning strategy that can adaptively focus on task-relevant information, maintaining both computational efficiency and task performance in diverse and dynamic robotic environments.

### 2.2.2 Model Pruning and Architecture Optimization

For model pruning, common strategies include weight-based methods (pruning small-magnitude parameters) (Frankle & Carbin, 2019; Han et al., 2016), learning-based methods like Movement Pruning (Sanh et al., 2020) that eliminate less active parameters during training, and output-sensitive

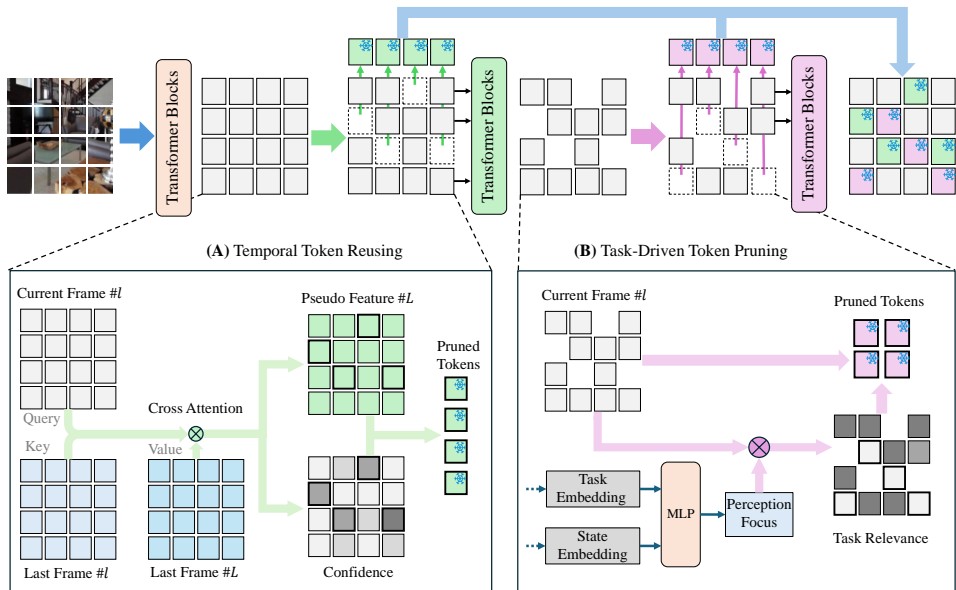

Figure 1: Overview of the Proposed Method: Given an input frame, the process consists of two pruning stages: (A) Temporal Token Reusing, where several tokens are identified as reusable based on the feature map of the previous frame. These tokens directly inherit the final features from the previous frame's extraction result, bypassing further processing. (B) Task-Driven Token Pruning, where tokens irrelevant to the task at hand are filtered out based on the current robot state and the task. These tokens are also excluded from subsequent feature extraction.

methods that prune based on prediction impact (Molchanov et al., 2019). These typically employ iterative pruning to balance efficiency and performance.

For attention acceleration, researchers have developed several paradigms to address quadratic complexity. Liu et al. (2021) introduces hierarchical local-window attention with shifted windows to capture global context efficiently in SwinT. Linear approximation methods (Shen et al., 2021; Han et al., 2023) reduce computation through projected attention scoring, while deformable attention mechanisms (Xia et al., 2022) reduce computational costs by attending to a dynamic subset of tokens.

## 3 METHOD

As discussed in section 2.2.1, existing approaches may prune critical information when the task-relevant content is not visually prominent, or retain irrelevant but salient regions, resulting in either reduced success rates or suboptimal efficiency.

In contrast, our method introduces task-driven token pruning, which dynamically adjusts to the specific needs of the task. This approach ensures that only the most relevant tokens are retained, improving processing efficiency without sacrificing success rates. As shown in Figure 1, our method employs two pruning modules to minimize redundant computation while preserving task-critical features. By selectively retaining tokens based on task relevance and temporal coherence, our system ensures efficient feature extraction, making it suitable for real-world robotic applications.

### 3.1 TEMPORAL TOKEN REUSING

In robotic environments, consecutive frames are often similar, with only small changes in the environment. Reprocessing redundant information can be computationally expensive. To address this, we propose a *temporal token reusing module*, which leverages feature consistency between adjacent frames. By reusing the final features from the previous frame, we approximate the current frame's output, avoiding unnecessary computation of redundant tokens. This approach can efficiently reduce computational overhead by focusing on newly observed elements.

Let $\mathbf{F}^{(t)} = \{\mathbf{f}_i^{(t)}\}_{i=1}^N$ denote the token features at a given Transformer layer for the current frame $t$, and $\mathbf{F}^{(t-1)} = \{\mathbf{f}_j^{(t-1)}\}_{j=1}^N$ the corresponding features for the previous frame. Let $\mathbf{Z}^{(t-1)} = \{\mathbf{z}_j^{(t-1)}\}_{j=1}^N$ represent the final output features of the previous frame.

We treat the current frame's tokens $\mathbf{F}^{(t)}$ as *queries*, the previous frame's features $\mathbf{F}^{(t-1)}$ as *keys*, and its final features $\mathbf{Z}^{(t-1)}$ as *values* in a cross-attention operation. The propagated feature for the $i$-th token in frame $t$ is computed as:

$$\hat{\mathbf{z}}_i^{(t)} = \text{CrossAttn}\left(\mathbf{f}_i^{(t)}, \mathbf{F}^{(t-1)}, \mathbf{Z}^{(t-1)}\right). \tag{1}$$

In this mechanism, each token in the current frame searches for similar tokens in the previous frame, using the attention scores between the queries (current frame tokens) and the keys (previous frame tokens) to determine which features from the previous frame are most relevant. The final feature for the $i$-th token in the current frame is then predicted based on the most relevant features from the previous frame, effectively bypassing the need for further computation of redundant tokens.

To determine whether a token $\mathbf{f}_i^{(t)}$ has a reliable match in the previous frame, we compute a confidence score $r_i$ based on three factors: (1) The **entropy** of attention weights $\mathbf{a}_i$, measuring the sharpness of the match. (2) The **cosine similarity** between predicted $\hat{\mathbf{z}}_i^{(t)}$ and the current-layer feature $\mathbf{f}_i^{(t)}$. (3) The **maximum attention weight** $\max(\mathbf{a}_i)$, indicating the strength of the most confident correspondence. These values are passed through a small MLP predictor $Q(\cdot)$ to produce a confidence score:

$$r_i = Q\left(\left[H(\mathbf{a}_i),\ \cos\left(\hat{\mathbf{z}}_i^{(t)}, \mathbf{f}_i^{(t)}\right),\ \max(\mathbf{a}_i)\right]\right), \quad r_i \in [0, 1]. \tag{2}$$

We then use a predefined threshold $\text{thr}_{\text{temp}}$ to categorize tokens:

$$\mathcal{I}_{\text{reuse}} = \{i \mid r_i \geq \text{thr}_{\text{temp}}\}, \tag{3}$$

$$\mathcal{I}_{\text{process}} = \{i \mid r_i < \text{thr}_{\text{temp}}\}. \tag{4}$$

At inference time, tokens in $\mathcal{I}_{\text{reuse}}$ are excluded from further Transformer computation. Their propagated features $\hat{\mathbf{z}}_i^{(t)}$ are directly passed to the final feature map. To preserve their influence, we enhance the remaining tokens by replacing $\mathbf{f}_i^{(t)}$ by $\hat{\mathbf{z}}_i^{(t)}$. This allows the retained tokens to integrate information from those that were confidently pruned, effectively transferring information extracted in previous frames.

### 3.2 Task-Driven Token Pruning

In robotic tasks, the relevance of visual information varies depending on the stage of the task. For instance, during the exploration phase of navigation, the robot needs to focus on detecting open spaces, obstacles, and potential pathways, while ignoring static, irrelevant background elements like walls. In contrast, when approaching the target, the robot must prioritize features related to the goal, such as target objects or landmarks, and minimize processing of peripheral elements.

To achieve this, we introduce a *task-driven token pruning module* that adaptively removes irrelevant tokens based on the robot's current state and task objective. By integrating a lightweight gating mechanism into the Transformer architecture, the model selectively predicts and retains task-relevant features and ensures efficient processing by focusing computation on the most important information at each stage of the task.

Let the input token features at a certain Transformer layer be denoted as $\{\mathbf{f}_i\}_{i=1}^N$, where $\mathbf{f}_i \in \mathbb{R}^d$ is the feature of the $i$-th token, and $N$ is the total number of tokens. Given the robot's current state $\mathbf{s}$ and task representation $\mathbf{t}$, we employ a lightweight focus predictor $P(\cdot)$ to estimate a perception focus:

$$\mathbf{c} = P(\mathbf{s}, \mathbf{t}), \tag{5}$$

where $\mathbf{c} \in \mathbb{R}^d$ encodes the semantic characteristics that the model should attend to under the current situation. For each token $\mathbf{f}_i$, we compute its cosine similarity to the perception focus $\mathbf{c}$:

$$\alpha_i = \cos(\mathbf{f}_i, \mathbf{c}) = \frac{\mathbf{f}_i^\top \mathbf{c}}{\|\mathbf{f}_i\| \cdot \|\mathbf{c}\|}. \tag{6}$$

We then use a predefined threshold $\text{thr}_{\text{task}}$ to categorize tokens:

$$\mathcal{I}_{\text{keep}} = \{i \mid \alpha_i \geq 1 - \text{thr}_{\text{task}}\}, \tag{7}$$

$$\mathcal{I}_{\text{drop}} = \{i \mid \alpha_i < 1 - \text{thr}_{\text{task}}\}. \tag{8}$$

Based on these similarity scores, we select tokens exceeding a predefined relevance threshold $\text{thr}_{\text{task}}$, which form the relevant token set $\mathcal{I}_{\text{keep}}$. The remaining tokens with scores below $\tau$ are considered irrelevant and form the set $\mathcal{I}_{\text{drop}}$.

At inference time, only the relevant tokens in $\mathcal{I}_{\text{keep}}$ contribute to the computation of subsequent Transformer blocks. The irrelevant tokens in $\mathcal{I}_{\text{drop}}$ are excluded from further processing; their features are directly passed on to the final feature map, bypassing any additional Transformer layers. This results in faster inference while maintaining the integrity of task-relevant computations.

## 3.3 IMPLEMENTATION DETAILS

**Temporal Token Reusing.** This module consists of a cross-attention block and a 2-layer MLP $Q$ for predicting the confidence score $r_i$ of temporal matches. It is pretrained using video sequences from the simulator. The training loss combines a regression term and a classification term:

$$\mathcal{L}_{\text{reuse}} = \text{MSE}(\hat{\mathbf{z}}_i, \mathbf{z}_i) + \text{CrossEntropy}\left(r_i, 1 - clip\left(\frac{\text{MSE}(\hat{\mathbf{z}}_i, \mathbf{z}_i)}{M}, [0, 1]\right)\right), \tag{9}$$

where the first term is a standard regression loss between the predicted feature $\hat{\mathbf{z}}_i$ and the ground-truth ViT feature $\mathbf{z}_i$. The second term supervises the confidence score $r_i$. To generate a target for this confidence, we transform the regression error into a pseudo-label by normalizing it with a dynamic threshold $M$. Based on our empirical observation of a sharp knee point in the sorted MSE distribution, we set $M$ to the value at this point, which is located using a knee-detection algorithm. This normalized error serves as a soft label to train the confidence prediction part.

**Task-Driven Token Pruning:** We train the focus predictor $P$ (a 2-layer MLP) using task-level supervision, where the agent is optimized to complete a downstream objective. The pruning decisions are guided by gradients derived from task-specific losses, allowing the model to learn which tokens are essential for task. This setup supports both reinforcement learning and imitation learning paradigms. In our implementation, we follow the settings of prior works (Majumdar et al., 2023) and adopt imitation learning using demonstration trajectories collected in simulation (Ramrakhya et al., 2022).

To enable end-to-end optimization, we adopt the *Straight-Through Estimator (STE)* (Bengio et al., 2013) to handle the non-differentiable token selection step. The binary retention mask $\tilde{m} = [\tilde{m}_1, \ldots, \tilde{m}_N]$ is generated based on the calculated similarity $\alpha_i$, and gradients are passed through by treating the sampling operation as the identity function during backpropagation.

In addition to the task loss, we introduce a sparsity loss to encourage the model to minimize the number of retained tokens. Specifically, we apply an $\ell_1$ penalty on the retention mask:

$$\mathcal{L}_{\text{sparse}} = \lambda \cdot \frac{1}{N} \sum_{i=1}^{N} \tilde{m}_i, \tag{10}$$

where $\lambda$ controls the trade-off between accuracy and efficiency. The final training objective is the sum of the task loss and the sparsity loss:

$$\mathcal{L}_{\text{prune}} = \mathcal{L}_{\text{task}} + \mathcal{L}_{\text{sparse}}. \tag{11}$$

This formulation allows the model to balance task performance with computational efficiency in a differentiable manner.

Table 1: Performance comparison of different methods on object navigation tasks, evaluated on both Orin (GPU) and Raspberry Pi (CPU) platforms. (sp) denotes the variant using only the Task-Driven Pruning, and (temp) denotes the variant using only the Temporal Reusing.

| | FPS | | GFLOPs | SPL(%) | SR(%) |
| | Orin (GPU) | Pi (CPU) | | | |
|---|---|---|---|---|---|
| Baseline | 37.4 | 0.69 | 8.99 | 22.03 | 52.10 |
| Baseline† | 48.6 | 1.17 | 6.42 | 13.76 | 35.25 |
| EViT (Liang et al., 2022) | 47.7 | 1.04 | 6.88 | 2.03 | 7.60 |
| ToMe (Bolya et al., 2023) | 40.9 | 1.20 | 6.02 | 12.30 | 32.10 |
| DynamicViT (Bolya et al., 2023) | 47.9 | 1.05 | 6.89 | 17.42 | 27.60 |
| Ours(sp) | 48.1 | 1.35 | 6.86 | 21.05 | 51.50 |
| Ours(temp) | 47.2 | 1.16 | 6.87 | 21.54 | 51.20 |
| Ours(full) | **56.4** | **1.95** | **5.00** | 21.46 | 50.05 |

## 4 EXPERIMENTS

### 4.1 EXPERIMENTAL SETUP

We evaluate our token reduction method on the ObjectNav task in the Habitat Challenge 2022, which involves navigating to six object categories (bed, chair, toilet, TV, sofa, plant) across 88 Gibson scenes, with 2000 validation episodes. Imitation learning is performed using 77k human demonstration trajectories collected via Habitat-Web (Ramrakhya et al., 2022).

Following prior works (Ramrakhya et al., 2023; Majumdar et al., 2023), we adopt an imitation learning paradigm to train the policy. The visual backbone of our model is a ViT, specifically the DeiT-Tiny (Touvron et al., 2021) model pretrained on ImageNet. To handle temporal dynamics, we append a recurrent neural network (RNN) after the transformer backbone, forming a ViT+RNN architecture for sequential decision-making. We insert the temporal token reusing module after the 3rd layer of the ViT and the task-driven token pruning module after the 6th layer, both with thresholds of 0.5 and a sparsity weight $\lambda = 0.1$.

We compare our method against the following baselines: (a) **Baseline**: A ViT+RNN model without pruning. (b) **Baseline†: A smaller baseline.** (c) **EViT** (Liang et al., 2022): A classic token pruning method that drops low-attention tokens based on their attention to the [CLS] token. (d) **ToMe** (Bolya et al., 2023): A token merging approach that fuses the most similar token pairs to reduce the number of tokens. (e) **DynamicViT** (Rao et al., 2021): A token pruning method that employs a learnable network to assign importance scores to tokens and subsequently discards those with the lowest scores. All methods are trained for 96,000 update steps on a system with eight NVIDIA RTX 4090 GPUs.

We report four metrics to evaluate both navigation performance and computational efficiency: **Frames Per Second (FPS)**: measuring inference speed. **GFLOPs**: measuring the amount of computation required for model inference. It indicates how many floating-point calculations are performed. **Success Rate (SR)**: the percentage of episodes where the agent reaches the target object within the time budget. **Success weighted by Path Length (SPL)**: which accounts for both success and path optimality. The shorter the path, the higher the SPL.

### 4.2 MAIN RESULTS

As shown in Table 1, our method consistently outperforms existing approaches in both efficiency and task performance. On the NVIDIA Orin platform, our full model achieves 56.4 FPS, representing a 50.8% improvement over the baseline and clearly surpassing other approaches. The advantage becomes even more pronounced on the resource-constrained Raspberry Pi platform, where our method reaches 1.95 FPS—nearly tripling the baseline speed (0.69 FPS). This scalability highlights the practicality of our approach for real-world robotic systems with limited computational resources.

In terms of computational cost, our method achieves the lowest complexity with 5.00 GFLOPs. Compared to alternative pruning-based approaches that remain above 6 GFLOPs and the baseline's 8.99 GFLOPs, this reduction translates directly into lower energy consumption and faster inference, reinforcing the efficiency of our design.

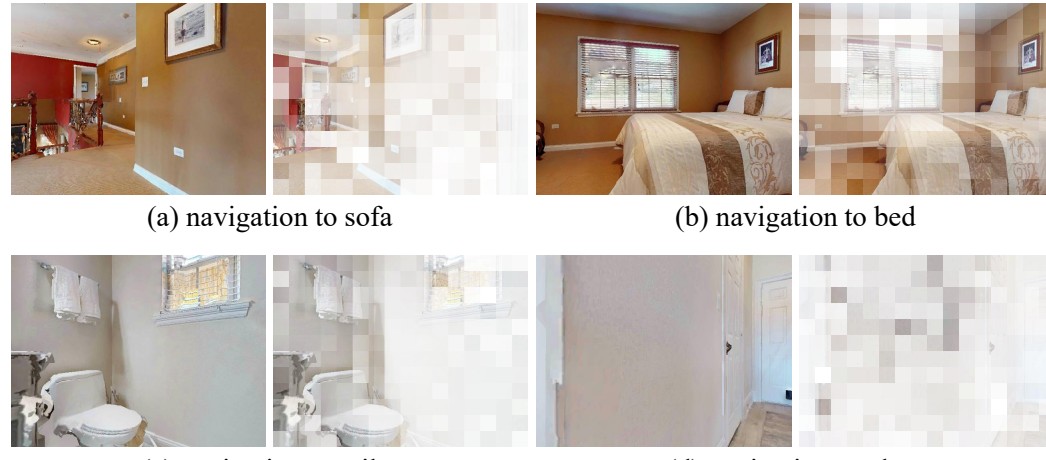

(a) navigation to sofa            (b) navigation to bed

(c) navigation to toilet           (d) navigation to plant

Figure 2: Visualization of token pruning. Each pair shows the original image on the left and the corresponding probability map on the right. More transparent areas indicate tokens that are more likely to be pruned.

Equally important, our method preserves navigation effectiveness. While several baselines exhibit severe degradation in Success Rate (SR) and Success weighted by Path Length (SPL), our method maintains 50.05% SR and 21.46% SPL. This corresponds to only a 2.05% and 0.57% absolute drop from the baseline, respectively—substantially smaller than the large declines observed in prior methods. Such results confirm that our acceleration does not come at the expense of reliable navigation behavior.

Overall, the results demonstrate that our method achieves scalable, hardware-agnostic acceleration through two complementary pruning strategies: temporal token reusing and task-driven token pruning. Unlike prior methods that rely on static or generic pruning, our design explicitly leverages temporal redundancy and task relevance to reduce computation while preserving navigation reliability, making it well-suited for real-world deployment across platforms with varying resources.

### 4.3 Token Pruning Visualization

To better illustrate the underlying mechanism, we visualize the pruning probability maps in Figure 2. The results show that the model consistently focuses on task-relevant regions (e.g., target objects, obstacles) while discarding uninformative background areas. This task-aware selection effectively reduces spatial redundancy and retains critical visual cues, which explains why our method can substantially increase inference speed without degrading navigation success rate.

### 4.4 Ablation Study

**Module Ablation.** The experimental results in Table 1 demonstrate how each component contributes to the overall performance. Using only the task-driven token pruning (Ours-sp) increases FPS to 48.1 (+28.6%) with minimal drop in success rate (51.5%), while temporal token reuse (Ours-temp) achieves 47.2 FPS with slightly higher SPL (21.54% vs. 21.05%), highlighting its benefit in preserving features across frames. The full model combines both, reaching 56.4 FPS (+51%) and maintaining success above 50%, confirming that both components are crucial for optimal performance.

**Threshold Ablation.** Figure 3 further investigates the effect of key thresholds and overall speed-performance trade-offs. Figure 3a shows that our method consistently outperforms all baselines across the entire SR–FPS curve, highlighting the effectiveness of task-driven token reduction in balancing efficiency and success rate.

Table 3b shows that the performance remains relatively stable across different temporal ($thr_{temp}$) and task-related ($thr_{task}$) thresholds, demonstrating the robustness of our approach. These results

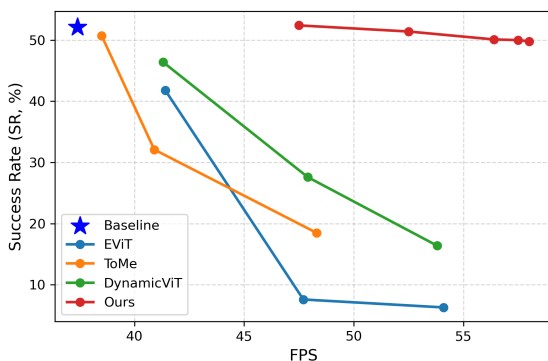

(a) SR–FPS trade-off curves on Orin. Our method consistently achieves a better balance between speed and success rate.

| thr$_{temp}$ | 0.3 | 0.4 | 0.5 | 0.7 | 0.9 |
|---|---|---|---|---|---|
| SR (%) | 49.0 | 50.0 | 50.1 | 51.1 | 51.8 |
| FPS | 57.1 | 56.7 | 56.4 | 49.3 | 41.6 |

| thr$_{task}$ | 0.3 | 0.4 | 0.5 | 0.7 | 0.9 |
|---|---|---|---|---|---|
| SR (%) | 42.8 | 50.1 | 50.1 | 50.3 | 51.4 |
| FPS | 57.3 | 57.0 | 56.4 | 49.7 | 38.7 |

(b) Effect of varying temporal (top) and task (bottom) thresholds.

Figure 3: Ablation study comparing the SR–FPS trade-off curves (left) and analyzing the effect of our method's key thresholds (right).

are obtained by directly adjusting the thresholds on the trained model without retraining, which also provides a simple way to control the trade-off between success rate and speed.

**Dynamic Scene Robustness.** Since temporal token reuse relies on inter-frame consistency, dynamic real-world conditions such as rapid camera motion (less spatial overlap) or sensor noise (less reusable infomation) can challenge its effectiveness. To isolate and analyze the behavior of it, we conduct controlled stress tests directly on the reuse module. Specifically, we simulate two perturbations: (i) **frame-skipping** (every 5th frame) to mimic rapid motion, and (ii) **Gaussian noise** ($\sigma = 0.1$) to simulate sensor degradation. We report two metrics: *MSE* measures the reconstruction error between features obtained via reuse and those recomputed from scratch, lower is better; *Reuse Rate* is the proportion of tokens reused without recomputation, higher indicates greater computational savings. Results in Table 2 show the module adaptively lowers reuse in harder cases ($39\% \rightarrow 33\%$) to maintain accuracy, with only a slight MSE increase (+0.012).

Table 2: Robustness of the temporal token reuse module under dynamic perturbations.

| Condition | MSE ↓ (Error) | Reuse Rate ↑ (Speed) |
|---|---|---|
| Normal | 0.177 | 39% |
| Frame-skipping | 0.184 | 38% |
| Gaussian noise | 0.189 | 33% |

Additional analysis and ablation studies are provided in Appendix A.

## 5 CONCLUSION

In this work, we introduced a biologically inspired token reduction framework to optimize the efficiency of Vision Transformers for real-time robotic applications. Our approach significantly improves inference speed, achieving a 1.5× to 3× acceleration without sacrificing essential navigation performance. Through extensive experiments on both high-performance GPU and resource-constrained CPU platforms (Jetson Orin and Raspberry Pi 4B), we demonstrated the scalability and effectiveness of our method across various hardware platforms.

The results highlight the potential of ViTs in robotics, particularly in tasks that require global context and spatial understanding. Our method enables practical deployment on edge devices, offering a viable solution for real-time robotic vision systems. Future work will focus on further enhancing the efficiency of ViTs through adaptive pruning strategies and exploring their application to more complex robotic tasks.

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

# A APPENDIX

## A.1 TEMPORAL TOKEN REUSE ANALYSIS

**Input Feature Ablation.**    To quantify the contribution of each input cue in the temporal token reuse decision, we conduct an ablation study. The three cues considered are: **attention entropy**, **maximum attention**, and **cosine similarity** between reconstructed and original token features.

We evaluate two metrics: (1) **Reuse MSE**, the mean squared error between reused token features and ground-truth features, indicating reuse accuracy; and (2) **Reuse Ratio**, the average percentage of tokens reused, indicating potential computational speed-up. Table 3 reports the results. Removing any single cue increases either the reuse error or reduces the reuse ratio, confirming that all three cues are necessary for accurate and stable token reuse.

Table 3: Ablation of input features for temporal token reuse.

| Confidence Cues Used | Reuse MSE ↓ (Error) | Reuse Ratio ↑ (Speed-up) |
|---|---|---|
| All cues (ours) | 0.177 | 39% |
| w/o entropy | 0.251 | 40% |
| w/o cosine similarity | 0.179 | 35% |
| w/o max attention | 0.262 | 39% |

## A.2 TASK-DRIVEN TOKEN PRUNING ANALYSIS

**Stage-wise Attention Behavior of Task-driven Token Pruning.**    We further examine the task-driven token pruning module's behavior across different navigation stages. Figure 4 visualizes the robot's attention focus at three stages of a navigation episode:

1. **Searching Stage**: The robot explores the environment, ignoring impassable areas such as walls and railings.

2. **Moving Stage**: Attention shifts to obstacles and the target object (e.g., bed or toilet), ensuring safe navigation.

3. **Reaching Stage**: The robot focuses almost entirely on the target, pruning irrelevant background tokens to achieve precise navigation.

This stage-wise analysis reveals that the pruning module exhibits a **phased behavior**, dynamically adjusting token retention according to the task context. Early in navigation, broader contextual tokens are preserved to support exploration, while later stages prioritize task-relevant tokens for accuracy and efficiency.

**Token Retention Across Different Goals.**    To evaluate the behavior of the task-driven token pruning module, we analyze token retention for different target objects. Due to simulator constraints, constructing identical trajectories with only the goal changed is infeasible, as this would alter the agent's exploration path and hidden state. To isolate the pruning module, we instead evaluate on static images with different goal instructions, treating each image as if the agent were starting fresh (hidden state reset to zero). The proportion of retained tokens per semantic category is measured and normalized within each goal.

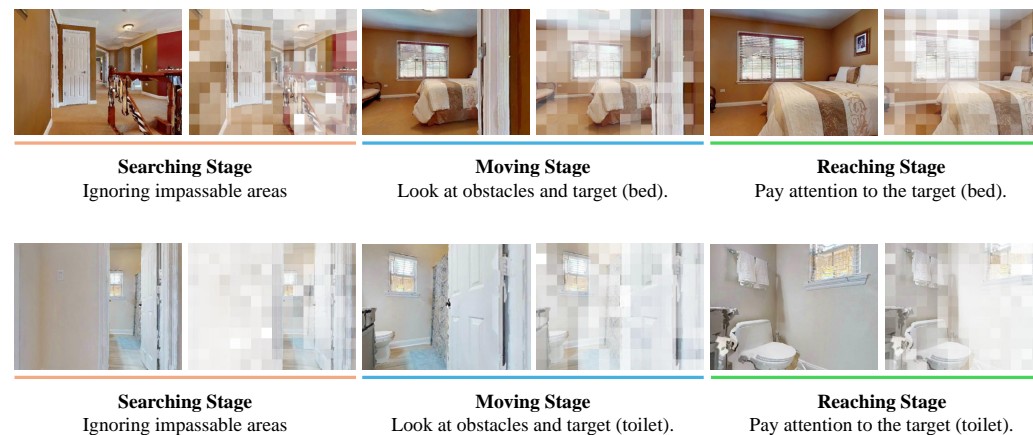

Figure 4: Stage-wise attention visualization during navigation. The robot progressively shifts focus from exploring the environment to attending to obstacles and finally concentrating on the target object for precise navigation.

Table 4 shows that the goal category consistently has the highest retention, indicating that the pruning module effectively focuses on task-relevant tokens while preserving necessary context for spatial understanding. Imperfect separation arises due to (i) object size (large objects occupy more area and are often attended to as navigation obstacles), (ii) contextual cues (non-target objects provide useful spatial context), and (iii) segmentation noise from the pretrained Mask2Former used for token labeling.

Table 4: Token retention per semantic category under different target goals.

| Goal | Chair Tokens | Bed Tokens | Plant Tokens | Other Tokens |
|---|---|---|---|---|
| Chair | 72% | 24% | 1.4% | 1.9% |
| Bed | 13% | 70% | 0.5% | 16% |
| Plant | 16% | 18% | 37% | 29% |

These results confirm that the pruning module reliably preserves task-relevant tokens while maintaining sufficient context for navigation.

**Target-Token Retention vs. Navigation Success.** To investigate the effect of target-token retention on navigation performance, we evaluate 200 validation images containing the target objects. For each image, we compute the per-image retention ratio as

$$\text{Retention} = \frac{\#\text{retained tokens of target object}}{\#\text{tokens of target object}},$$

and average across images to mitigate object size bias.

Table 5 reports layer-wise retention and navigation performance for our task-driven token pruning module and two baseline methods (DynamicViT and EViT). Our module prunes only at layer 6, maintaining high retention of task-relevant tokens, whereas DynamicViT and EViT progressively discard goal-relevant tokens across layers. Higher layer-wise retention consistently corresponds to better success rate, demonstrating that preserving task-relevant tokens is critical for navigation.

These results indicate that task-agnostic pruning strategies, such as DynamicViT, fail to retain sufficient goal-relevant tokens, leading to lower navigation success. In contrast, our task-driven token pruning effectively maintains high retention at key layers, explaining its superior performance.

Table 5: Layer-wise target-token retention and navigation performance.

| Method | L3 (%) | L6 (%) | L9 (%) | Success Rate (%) | FPS |
|---|---|---|---|---|---|
| Ours | – | 74.0 | – | 51.5 | 47.2 |
| DynamicViT | 52.6 | 28.2 | 12.3 | 27.6 | 47.9 |
| EViT | 61.3 | 26.1 | 8.6 | 7.6 | 47.7 |

A.3 IMPACT OF MODULE INSERTION LAYERS

We investigate the impact of the insertion positions for the Temporal Reuse and Task-Driven Pruning modules. The quantitative results are presented in Table 6.

Moving the **Temporal Reuse module** to a deeper layer (L6) results in a noticeable reduction in inference speed with negligible performance gain. This indicates that exploiting temporal redundancy in early layers is crucial for maximizing computational savings, as it prevents redundant computations for a larger portion of the network depth.

Conversely, positioning the **Task-Driven Pruning module** at an earlier layer (L3) leads to a drastic collapse in navigation performance, with the Success Rate dropping significantly to 35.10%. This sharp decline confirms that shallow features lack the high-level semantic representation required for making accurate task-relevance decisions.

These empirical results validate that our default configuration (Reuse at L3, Pruning at L6) offers the optimal balance between computational efficiency and navigation success.

Table 6: Ablation study on the insertion layers of token reduction modules. We evaluate the impact of placing the Temporal Reuse and Task-Driven Pruning modules at different layers. The default configuration (gray row) achieves the best trade-off.

| Reuse Layer | Prune Layer | FPS (Orin) | Success Rate (%) |
|---|---|---|---|
| L3 | L3 | 57.3 | 35.10 |
| L6 | L6 | 54.6 | 50.20 |
| L3 (Default) | L6 (Default) | 56.4 | 50.05 |

