# OpenReview forum: "Seeing Like Humans: Task-Driven Token Reduction for Accelerated ViT in Robotic Navigation"
_ICLR.cc/2026/Conference — Submitted to ICLR 2026_

### Official Review · Reviewer_kYmX · 2025-10-31

**Soundness:** 3
**Presentation:** 3
**Contribution:** 3
**Rating:** 6
**Confidence:** 2

**Summary:**

The paper proposes a human-like, biological-inspired token reduction method for using ViTs in robotic navigation that accelerates inference. The authors propose to reuse temporal token reusing that reuses previous frame via cross-attention and skips recomputing redundant tokens. The authors also propose task-driven token pruning, which predicts task relevance from the robot’s current state and goal and drops tokens that are irrelevant to the final objective. The experiment results show up to ~1.5× speedup on Orin and nearly 3× speedup on Raspberry Pi with only a small drop in success rate and SPL compared to full baseline methods.

**Strengths:**

1. From a perspective of pruning to what human sees is quite interesting and novel. The cross-attention between the current frame’s tokens and the previous frame’s final features, plus a learned confidence score that blends attention entropy, cosine similarity, and max attention to decide which tokens can skip recomputation, is quite similar to what human sees through time.
2. The policy-aware pruning is also an intuitive design for this specific robot navigation task.
3. The paper provides qualitative and quantitative interpretability, where the paper provides visualizations that show that the model keeps tokens around the goal object and navigable path while discarding irrelevant background.
4. The experiment results show acceleration while maintaining an acceptable drop compared to the full baseline method. And the added modules are lightweight.

**Weaknesses:**

1. The current evaluation seems to be narrowly focused on ObjectNav in Habitat with imitation-learned policies. Would the same pruning technique also work for other embodied tasks, such as manipulation, dynamic obstacle avoidance, and outdoor navigation?
2. The temporal reuse block assumes adjacent frames are similar enough that features can be carried forward. Will the solution still work under tasks involving fast egomotion, motion blur, or abrupt viewpoint shifts that are common on robots in the wild?
3. The baselines (EViT, DynamicViT, ToMe) are standard token pruning methods from vision, as they do not have task-aware pruning for the task paper's focus. Will this create a bias in the comparison in the experiment?

**Questions:**

Please see the weakness section.

---

> ### Author Response · Authors · 2025-11-24
>
> We thank the reviewer for the positive assessment and for finding our biologically-inspired approach **"interesting and novel"** and our design **"intuitive"**. We appreciate your recognition of the interpretability and efficiency gains.
>
> ## W1
>
> > "The current evaluation seems to be narrowly focused on ObjectNav in Habitat with imitation-learned policies. Would the same pruning technique also work for other embodied tasks, such as manipulation, dynamic obstacle avoidance, and outdoor navigation?"
>
> We validated our method on the **Meta-World** `reach-v3` manipulation task. As shown below, our method achieves a 2.3x speedup on Raspberry Pi compared to the baseline while maintaining a comparable Success Rate. This demonstrates that our framework generalizes well beyond ObjectNav.
>
> | Method   | FPS (Raspberry Pi) | Success Rate (%) |
> |:-------- |:------------------:|:----------------:|
> | Baseline | 0.75               | 60%              |
> | ToMe     | 1.33               | 51%              |
> | Ours     | 1.72               | 56%              |
>
> ## W2
>
> > "The temporal reuse block assumes adjacent frames are similar enough that features can be carried forward. Will the solution still work under tasks involving fast egomotion, motion blur, or abrupt viewpoint shifts that are common on robots in the wild?"
>
> This is an important question. While simulating extreme "wild" conditions in standard Habitat simulator is challenging, we conducted a controlled experiment to evaluate the robustness of our Temporal Reuse Module under "Fast Motion" (Frame-skipping) and "Sensor Degradation" (Gaussian noise) in **Table 2 of the main paper**.
>
> As summarized below, the results demonstrate that our Temporal Reuse Module behaves **adaptively**. Under challenging conditions (Frame-skip/Noise), the model automatically **reduces the Reuse Rate** from 39% to 33% to prevent the Feature Error (MSE) from spiking. This confirms that the model intelligently decides *when* to reuse and when to recompute, ensuring robustness against abrupt changes.
>
> | Condition                                   | MSE ↓ (Feature Error) | Reuse Rate ↑ (Speed) |
> |:------------------------------------------- |:---------------------:|:--------------------:|
> | Normal (Standard Task)                      | 0.177                 | 39%                  |
> | Frame-skip (Simulating Fast Motion)         | 0.184                 | 38%                  |
> | Gaussian Noise (Simulating Sensor Noise)    | 0.195                 | 33%                  |
>
> ## W3
>
> > "The baselines (EViT, DynamicViT, ToMe) are standard token pruning methods from vision, as they do not have task-aware pruning for the task paper's focus. Will this create a bias in the comparison in the experiment?"
>
> We respectfully clarify that this difference highlights our **core contribution** rather than an experimental bias.
>
> Most existing pruning methods, including recent MLLM-based ones (e.g., FastV[1], VisionZip[2]), are largely task-agnostic. Our work bridges this gap by introducing "Task-Driven Token Pruning," which we argue is essential for embodied AI.
>
>
> **References**
>
> [1] Chen et al. "An Image is Worth 1/2 Tokens After Layer 2: Plug-and-Play Inference Acceleration for Large Vision-Language Models." ECCV 2024.
>
> [2] Yang et al. "Visionzip: Longer is better but not necessary in vision language models." CVPR 2025.

---

> > ### Comment · Reviewer_kYmX · 2025-11-26
> >
> > I appreciate the author's rebuttal response. My concerns have been solved. I am happy to maintain the original rating and continue to recommend accepting the paper.

---

### Official Review · Reviewer_srF1 · 2025-11-01

**Soundness:** 2
**Presentation:** 3
**Contribution:** 2
**Rating:** 4
**Confidence:** 5

**Summary:**

This paper aims at addressing the quadratic computational cost of ViTs, which limits the real-time use on edge devices. Specifically, the authors draw inspiration from human vision and introduce two key ideas: (1) task-driven token pruning that selectively processes image regions relevant to the current task/state, while discarding irrelevant visual tokens, and (2) temporal token reuse that reuses stable visual features across frames to avoid redundant computation in sequential video input. The experiments show that the method achieves up to 1.5×–3× speedup with minimal loss in navigation performance compared to standard ViTs.

**Strengths:**

- The paper directly focuses on embodied AI and practical robotics settings, which sounds useful. The discussion of efficiency constraints and control-loop implications also make sense to me.
- The experiments are conducted on multiple hardware platforms, such as Jetson Orin and Raspberry Pi 4B, which demonstrate meaningful real-world value.
- Based on the provided table, the proposed method achieves significant acceleration (up to 3×) without sacrificing navigation success rates close to baseline models. The results look promising.
- The paper is well written and easy to follow. The figures are well-plotted and informative which can make readers quickly understand the core ideas.

**Weaknesses:**

- My major concern is that all of the experiments are limited to ObjectNav in Habitat, which show limited diversity on the target task. Such experiments make it questionable that how the proposed method can be generalized to other manipulation or multi-object reasoning tasks.
- Another concern is that the authors only provide FPS and success rate, which are too narrow to evaluate on the target task. The authors should also provide the evaluation results on perception quality, such as visual feature fidelity and localization accuracy.
- The authors only compare with some pruning-based and merging-based methods (e.g., EViT, ToMe, DynamicViT) but lack of the comparisons with some other ViT acceleration approaches, such as some linear attention methods (e.g., Performers [1]) and compression-based methods (e.g., MobileViT [2]).
- The hyperparameters of "temporal token reuse threshold" and "task-driven pruning threshold" directly determine how aggressively tokens are reused or pruned, which could very sensitive to the balance between computational savings and perceptual fidelity. However, the authors naively use 0.5 throughout the paper without further ablation. It would be insightful to see how these hyperparamters affect the model performance.


[1] "Rethinking Attention with Performers", ICLR 2021
[2] "MobileViT: Light-weight, General-purpose, and Mobile-friendly Vision Transformer", ICLR 2022

**Questions:**

- Could the authors provide the evaluation on some other tasks with different manipulations or on multi-object reasoning tasks?
- Could the authors provide the scores of visual feature fidelity and localization accuracy in the table?
- Could the authors compare more related acceleration models/approaches?
- Could the authors provide the ablation study of the hyperparameters, such as "temporal token reuse threshold" and "task-driven pruning threshold"?

---

> ### Author Response · Authors · 2025-11-24
> **Part I**
>
> We thank the reviewer for recognizing the **"meaningful real-world value"** of our work and **"significant acceleration (up to 3×)"** of our work with minimal performance loss. We appreciate your constructive feedback on generalization and evaluation metrics.
>
> ## W1
>
> > "My major concern is that all of the experiments are limited to ObjectNav in Habitat, which show limited diversity on the target task. Such experiments make it questionable that how the proposed method can be generalized to other manipulation or multi-object reasoning tasks."
>
> We validated our method on the Meta-World `reach-v3` manipulation task. As shown below, our method achieves a 2.3x speedup on Raspberry Pi compared to the baseline with a comparable Success Rate. This confirms that our method effectively generalizes to manipulation settings.
>
> | Method   | FPS (Raspberry Pi) | Success Rate (%) |
> |:-------- |:------------------:|:----------------:|
> | Baseline | 0.75               | 60%              |
> | ToMe     | 1.33               | 51%              |
> | Ours     | 1.72               | 56%              |
>
> ## W2
>
> > "Another concern is that the authors only provide FPS and success rate, which are too narrow to evaluate on the target task. The authors should also provide the evaluation results on perception quality, such as visual feature fidelity and localization accuracy."
>
> Since our model is trained end-to-end for navigation actions (without auxiliary heads), it does not produce reconstruction or localization outputs. So, we can not calculate quantitative "visual feature fidelity" or "localization accuracy" scores directly.
>
> However, we performed a **"Token Retention Analysis"** (detailed in **the Appendix A2**) to verify if the model focuses on the correct regions. We measured the percentage of retained tokens belonging to the Goal Category vs. Distractor Categories.
>
> For your convenience, we summarize the results below. The Goal Category consistently has the highest retention rate. This quantitatively confirms that our Task-Driven Pruning module "localizes" and preserves task-relevant tokens while discarding irrelevancies.
>
> | Goal Category | Retained "Chair" Tokens | Retained "Bed" Tokens | Retained "Plant" Tokens | Other Tokens |
> |:------------- |:-----------------------:|:---------------------:|:-----------------------:|:------------:|
> | **Chair**     | **72%**                 | 24%                   | 1.4%                    | 1.9%         |
> | **Bed**       | 13%                     | **70%**               | 0.5%                    | 16%          |
> | **Plant**     | 16%                     | 18%                   | **37%**                 | 29%          |
>
> *Note: The retention is not 100% exclusive due to: (1) Large obstacles (e.g., beds) often need to be retained for collision avoidance; (2) Contextual cues essential for spatial reasoning.*
>
> ## W3
>
> > "The authors only compare with some pruning-based and merging-based methods (e.g., EViT, ToMe, DynamicViT) but lack of the comparisons with some other ViT acceleration approaches, such as some linear attention methods (e.g., Performers [1]) and compression-based methods (e.g., MobileViT [2])."
>
> We respectfully clarify that our method is **orthogonal** to architectural modifications like Linear Attention or MobileViT. These approaches optimize attention complexity or parameter structure, whereas our work focuses on reducing input data redundancy.
>
> To ensure a fair, "apples-to-apples" comparison, we followed standard evaluation protocols [1,2] by benchmarking primarily against representative token pruning/merging methods like ToMe and DynamicViT. This allows us to isolate the contribution of our novel selection mechanism without confounding results with architectural differences.

---

> > ### Author Response · Authors · 2025-11-24
> > **Part II**
> >
> > ## W4
> >
> > > "The hyperparameters of "temporal token reuse threshold" and "task-driven pruning threshold" directly determine how aggressively tokens are reused or pruned, which could very sensitive to the balance between computational savings and perceptual fidelity. However, the authors naively use 0.5 throughout the paper without further ablation. It would be insightful to see how these hyperparamters affect the model performance."
> >
> > We respectfully point out that **Figure 3 in the main paper** already presents the trade-off constructed by adjusting these thresholds. This demonstrates our method's superior efficiency-accuracy balance.
> >
> > * **Choice of 0.5:** We clarify that the default value (0.5) was selected as a **representative operating point** to facilitate a standardized comparison with baselines in the main tables, rather than implying a single optimal setting.
> > * **Expanded Ablation:** To further address your concern, we have expanded the value range in Figure 3b below. The results confirm that the thresholds act as tunable "control knobs": increasing them yields higher SR but gradually lowers FPS.
> >
> > **(a) Ablation on Temporal Reuse Threshold ($thr_{temp}$)**
> >
> > | Threshold    | 0.1  | 0.3  | 0.4  | **0.5 (default)** | 0.6  | 0.7  | 0.9  |
> > |:------------ |:----:|:----:|:----:|:-----------------:|:----:|:----:|:----:|
> > | Success Rate | 42.6 | 49.0 | 50.0 | **50.1**          | 50.6 | 51.1 | 51.8 |
> > | FPS          | 57.5 | 57.1 | 56.7 | **56.4**          | 55.1 | 49.3 | 41.6 |
> >
> > **(b) Ablation on Task-Driven Pruning Threshold ($thr_{task}$)**
> >
> > | Threshold    | 0.1  | 0.3  | 0.4  | **0.5 (default)** | 0.6  | 0.7  | 0.9  |
> > |:------------ |:----:|:----:|:----:|:-----------------:|:----:|:----:|:----:|
> > | Success Rate | 17.7 | 42.8 | 50.1 | **50.1**          | 50.2 | 50.3 | 51.4 |
> > | FPS          | 58.2 | 57.3 | 57.0 | **56.4**          | 53.6 | 49.7 | 38.7 |
> >
> > These results have been included in Table 3b of the updated paper.
> >
> > **References**
> >
> > [1] Bolya et al. "Token Merging: Your ViT But Faster." ICLR 2023.
> >
> > [2] Rao et al. "Dynamicvit: Efficient vision transformers with dynamic token sparsification."  NeurIPS 2021.

---

### Official Review · Reviewer_Fuvt · 2025-11-01

**Soundness:** 3
**Presentation:** 3
**Contribution:** 3
**Rating:** 4
**Confidence:** 4

**Summary:**

This paper proposes a token reduction framework to optimize effficiency of Vision Transformers (ViTs) for robotic navigation. The authors propose a biologically-inspired token reduction framework with two components: (1) a temporal token reuse module that identifies and reuses stable visual features across consecutive frames to avoid redundant computation, and (2) a task-driven token pruning module that selectively retains tokens relevant to the current navigation task and robot state while discarding irrelevant regions.

The method is evaluated on object navigation tasks in the Habitat simulator using a ViT+RNN architecture with DeiT-Tiny as the visual backbone. The temporal reuse module is inserted after the 3rd ViT layer and uses cross-attention to propagate features from the previous frame, while the task-driven pruning module is placed after the 6th layer and uses a lightweight MLP to predict a "perception focus" based on robot state and task. Unlike prior token reduction methods that use fixed pruning ratios, this approach dynamically adapts token selection based on temporal redundancy and task requirements, allowing a single model to generalize across multiple navigation goals without retraining.

The results demonstrate substantial speedups: 1.5× on Jetson Orin GPU (56.4 FPS vs. 37.4 FPS baseline) and nearly 3× on Raspberry Pi 4B CPU (1.95 FPS vs. 0.69 FPS baseline), while maintaining navigation performance with only minor drops in Success Rate (50.05% vs. 52.10%) and SPL (21.46% vs. 22.03%). The method reduces computational cost to 5.00 GFLOPs compared to the baseline's 8.99 GFLOPs.

**Strengths:**

1. The paper is well-written and the approach is simple, intuitive, and works reliably
2. Compared to prior toke-pruning methods the proposed approach performs the best by being efficient in using less FLOPs and maintaining high success rates as presented in main results which is quite promising.
3. The analysis and ablations presented in section 4.3 are  well thought through and demonstrate benefits of using the proposed approach over prior methods. It also clearly demonstrates that even after trading off compute through pruning the method maintains high success rates.

**Weaknesses:**

1. The choice of adding the token reuse module at 3rd layer of ViT and token pruning module on 6th layer of the ViT are not very well motivated and explained in the paper. Can authors elaborate on how this decision was made? Also I believe ablating the choice of layers for each of these modules and adding those results to the paper will be quite valuable.
2. Similarly the choice of threshold values and lambda are not very well supported through relevant ablations. It would be great if authors can add justification for them and add supporting experiments.
3. The paper also doesn’t mention how the output tokens from ViT are converted and passed to the RNN policy. I am curious why authors choose a RNN for policy model instead of using a transformer and I would be interested in seeing same comparison of all methods presented in table 1 with a ViT+Transformer policy. I am concerned that if the ViT outputs are compressed using a spatial compression layer for RNN input that could be causing issues with other methods like ToME and DynamicViT. Operating directly on visual encoder output the transformer policy for navigation could give more insights into this. However this will incur more FLOPs as now we need to maintain K frames in transformer context for reasonable performance as there is no hidden states of past interactions.

**Questions:**

The contributions in the paper show meaningful performance and efficiency improvements. There are a few decision decisions for which ablations are missing or not justified due which I recommend a borderline reject. I am happy to increase my ratings if authors can address the concerns I mentioned and add supporting experiments

---

> ### Author Response · Authors · 2025-11-24
> **Part I**
>
> We thank the reviewer for the constructive feedback and for recognizing that our approach is **"simple, intuitive, and works reliably"** and that our analysis is **"well thought through"**. We are encouraged by your assessment that our method **"performs the best"** among prior pruning methods. We particularly appreciate your willingness to reconsider the rating based on our clarifications.
>
> ## W1
>
> > "The choice of adding the token reuse module at 3rd layer of ViT and token pruning module on 6th layer of the ViT are not very well motivated and explained in the paper. Can authors elaborate on how this decision was made? Also I believe ablating the choice of layers for each of these modules and adding those results to the paper will be quite valuable."
>
> * **Alignment with Literature:** Layers 3, 6, and 9 naturally divide the 12-layer ViT into four equal stages. These are the standard positions for inserting pruning modules in existing methods (e.g., **DynamicViT [1]**, **SViT [2]**). We adopted these standard locations to ensure a fair and consistent comparison with prior works.
> * **Module-Specific Rationale:** The specific assignment of **Reuse at Layer 3** and **Pruning at Layer 6** is motivated by the hierarchical nature of visual features:
>   * **Layer 3 (Temporal Reuse):** The core mechanism of this module is to compare tokens between the current and previous frames to identify reusable redundant parts. This matching process inherently relies on **detailed visual information** rather than abstract semantics. Since shallow layers in ViT specialize in encoding these low-level features, Layer 3 is a reasonalbe placement.
>   * **Layer 6 (Task-Driven Pruning):** This module requires **high-level semantic understanding** to determine task relevance. Such semantic information is typically well-formed in middle/deep layers, making Layer 6 an ideal insertion point.
> * **Ablation Verification:** We conducted the requested ablation. As shown in the table below, moving the Reuse module deeper to Layer 6 diminishes speed gains as it fails to exploit redundancy in the early stages. Conversely, moving the Pruning module earlier to Layer 3 significantly degrades the Success Rate, as the shallow features lack sufficient semantic information for accurate task-relevant decisions. This empirically confirms that our default configuration is a balanced choice.
>
> | Reuse Layer  | Prune Layer  | FPS (Orin) | Success Rate (%) |
> |:------------:|:------------:|:----------:|:----------------:|
> | L3 (default) | L6 (default) | 56.4       | 50.05            |
> | L6           | L6           | 54.6       | 50.20            |
> | L3           | L3           | 57.3       | 35.10            |
>
> These results have been included in Appendix A.3.1 of the updated paper.
>
> ## W2
>
> > "Similarly the choice of threshold values and lambda are not very well supported through relevant ablations. It would be great if authors can add justification for them and add supporting experiments."
>
> We provided an initial analysis in **Figure 3b** of the main paper showing the influence of the thresholds. To further support our choice, we expanded the ablation range for both the **Temporal Reuse Threshold** ($thr_{temp}$) and the **Task-Driven Pruning Threshold** ($thr_{task}$) as shown bellow. The results demonstrate a clear **trade-off**.
> As the thresholds decrease, the system becomes more aggressive in reduction, leading to higher FPS but lower Success Rate.
>
> This confirms that the threshold parameters effectively control the balance. We clarify that our default value (0.5) was selected as a **representative operating point** to facilitate standardized comparisons with baselines, rather than implying a single optimal setting.
>
> **(a) Ablation on Temporal Reuse Threshold ($thr_{temp}$)**
>
> | Threshold    | 0.1  | 0.3  | 0.4  | **0.5 (Ours)** | 0.6  | 0.7  | 0.9  |
> |:------------ |:----:|:----:|:----:|:--------------:|:----:|:----:|:----:|
> | Success Rate | 42.6 | 49.0 | 50.0 | **50.1**       | 50.6 | 51.1 | 51.8 |
> | FPS          | 57.5 | 57.1 | 56.7 | **56.4**       | 55.1 | 49.3 | 41.6 |
>
> **(b) Ablation on Task-Driven Pruning Threshold ($thr_{task}$)**
>
> | Threshold    | 0.1  | 0.3  | 0.4  | **0.5 (Ours)** | 0.6  | 0.7  | 0.9  |
> |:------------ |:----:|:----:|:----:|:--------------:|:----:|:----:|:----:|
> | Success Rate | 17.7 | 42.8 | 50.1 | **50.1**       | 50.2 | 50.3 | 51.4 |
> | FPS          | 58.2 | 57.3 | 57.0 | **56.4**       | 53.6 | 49.7 | 38.7 |
>
> These results have been included in Table 3b of the updated paper.
>
> $\lambda$ acts as a penalty to distinguish task-relevant tokens from redundancy. We observe that the learned relative ranking is robust to small variations in $\lambda$. Therefore, precise control relies on the pruning threshold at inference, rendering extensive hyperparameter search for $\lambda$ unnecessary.

---

> > ### Author Response · Authors · 2025-11-24
> > **Part II**
> >
> > ## W3
> >
> > > "The paper also doesn’t mention how the output tokens from ViT are converted and passed to the RNN policy. I am curious why authors choose a RNN for policy model instead of using a transformer and I would be interested in seeing same comparison of all methods presented in table 1 with a ViT+Transformer policy. I am concerned that if the ViT outputs are compressed using a spatial compression layer for RNN input that could be causing issues with other methods like ToME and DynamicViT. Operating directly on visual encoder output the transformer policy for navigation could give more insights into this. However this will incur more FLOPs as now we need to maintain K frames in transformer context for reasonable performance as there is no hidden states of past interactions."
> >
> > * **Choice of RNN:** We selected the RNN-based policy because it is a standard, representative, and concise baseline in the object navigation task (e.g., VC-1 [3]), which achieves SOTA performance. Adopting this established architecture ensures that our comparisons focus strictly on token reduction efficiency rather than policy architectural changes.
> >
> > * **Transformer Policy:** Migrating to a full Transformer policy is a valid and interesting direction. However, since our task-driven module utilizes the RNN hidden state, adapting it to a Transformer architecture requires network redesign and extensive hyperparameter tuning. Due to the limited rebuttal timeframe, we could not complete this full migration.
> >
> > * **Fairness Mechanism:** Regarding the compression layer, it is also employed in VC-1[3] to compress the visual features output by ViT. We clarify that it exhibits **no architectural bias**. We follow standard practices in token pruning (e.g., SViT[2]) to generate dense feature maps: rather than masking pruned tokens with zeros, we directly fill the pruned locations with their **feature values before pruning** (bypassing subsequent blocks), while retained tokens are processed normally. Specifically, for ToMe, we copy the merged token and place it back in the positions of the tokens before merging. Finally, we reconstruct the full feature map. This ensures the compression layer always processes a spatially complete map with valid features, treating all methods **equally**.
> >
> > * **Transformer-based Compression:** To further address your concern, we replaced the CNN-based compression with a **Transformer-based compression layer** (aggregating features into a single token) before the RNN.
> >
> >   The Baseline's Success Rate dropped from **52%** to **32%**, ToMe dropped from **31%** to **26%** under the same pruning ratio as the main paper.
> >   This confirms that the standard CNN-based compression is better for this architecture, indicating that the compression layer does not hinder these methods.
> >
> > **References:**
> >
> > [1] Rao et al., "DynamicViT: Efficient Vision Transformers with Dynamic Token Sparsification", NeurIPS 2021.
> >
> > [2] Liu et al. "Revisiting token pruning for object detection and instance segmentation." WACV 2024.
> >
> > [3] Majumdar et al. "Where are we in the search for an artificial visual cortex for embodied intelligence?." NeurIPS 2023.

---

> > > ### Comment · Reviewer_Fuvt · 2025-11-25
> > > **Response to authors**
> > >
> > > Thanks for the detailed response and additional results. Overall I think this is a good paper with interesting experiments. The response from authors have addressed my concerns so I updated my rating. I recommend authors to add the additional details and results discussed to the final paper.

---

### Official Review · Reviewer_qd63 · 2025-11-02

**Soundness:** 3
**Presentation:** 2
**Contribution:** 2
**Rating:** 4
**Confidence:** 3

**Summary:**

This paper proposes a framework to improve the efficiency of Vision Transformers (ViTs) for robotic navigation. The method introduces:

1) Temporal Token Reusing : cross-frame attention to reuse redundant features across consecutive frames.

2) Task-Driven Token Pruning : dynamically removing spatial tokens irrelevant to the robot’s current task or goal.

The authors evaluate the approach on Habitat’s ObjectNav benchmark, showing speedups on Jetson Orin (GPU) and Raspberry Pi 4B (CPU), with small drops in navigation success.

**Strengths:**

Making visual robotic policies run on-device in real time in embodied settings is an important problem to solve with a general solution that doesn’t assume too much about the problem setting. The approach here shows speed gain, outperforming other pruning baselines that were studied on the task.

**Weaknesses:**

1) Results are shown only for one task with a specific assumption (the environment will be largely static for this method to work), the empirical evidence would be much stronger if the authors show applicability of method at least on manipulation and Imagenav tasks - both have similar assumptions as the task in the paper (static environment, small localised motions).

2) Even on the single task that is used to study the method, SR/SPL are below the baseline (albeit slightly).

3) I would want to see results on a baseline that is essentially the temporal version of ToMe (trivial modification to reuse tokens across frames if they satisfy the same closeness criteria as that applied in ToMe at the single frame level currently) for fairness

4) I would also compare performance to a smaller model that is flop matched to the proposed method to check if the pruning method proposed outperforms that at least.

5) ToMe is a training free approach without much overhead, so the authors should report how much is the overhead from their approach because they have to train two stages on top of the baseline - since if it is comparable compute to training the baseline itself then that is not very meaningful

6) I see no change in performance in table 3b for temporal ablations when varying the threshold - maybe we need to ablate the range further to get some insights from this ablation?

7) Writing needs to be improved to make the paper more self contained and to make the experimental section easier to parse. The experimental section does not define (sp) and (temp) used in the main results table anywhere. There should also be a description of what the different task subsets (base/long) are actually testing and how they differ.

**Questions:**

Listed above.

---

> ### Author Response · Authors · 2025-11-24
> **Part I**
>
> We thank the reviewer for the detailed review and for recognizing that real-time on-device policy is an **"important problem to solve"**. We also appreciate your acknowledgment that our approach is a **"general solution"** that **"outperforming other pruning baselines."** We address your comments below.
>
> ## W1
>
> > "Results are shown only for one task with a specific assumption (the environment will be largely static for this method to work), the empirical evidence would be much stronger if the authors show applicability of method at least on manipulation and Imagenav tasks - both have similar assumptions as the task in the paper (static environment, small localised motions)."
>
> Following your suggestion, we validated our method on the **Meta-World** `reach-v3` task. As shown in the table below, our method achieves a 2.3x speedup on Raspberry Pi compared to the baseline while maintaining a comparable Success Rate. This demonstrates that our framework generalizes well to manipulation settings.
>
> | Method   | FPS (Raspberry Pi) | Success Rate (%) |
> |:-------- |:------------------:|:----------------:|
> | Baseline | 0.75               | 60%              |
> | ToMe     | 1.33               | 51%              |
> | Ours     | 1.72               | 56%              |
>
> ## W2
>
> > "Even on the single task that is used to study the method, SR/SPL are below the baseline (albeit slightly)."
>
> We acknowledge the minor success rate drop. However, we emphasize that such a trade-off is a **common phenomenon** in token reduction research. While maintaining lossless performance is the ideal, achieving significant acceleration typically incurs a minor cost with current technologies. Therefore, the practical research goal is to push the Pareto frontier to optimize this trade-off. Our contribution remains robust for two key reasons:
>
> 1. **Superior Trade-off:** As shown in Figure 3, our method achieves a **better efficiency-accuracy trade-off curve** compared to other methods. This means that for a given speedup, our method preserves more task-relevant information.
> 2. **Robotic Value:** Specifically in the embodied context, exchanging a negligible SR drop for a **3x speedup** is highly favorable. The system-level benefit of high-frequency control enables safer and more responsive navigation, outweighing the minor drop in static metrics.
>
> ## W3
>
> > "I would want to see results on a baseline that is essentially the temporal version of ToMe (trivial modification to reuse tokens across frames if they satisfy the same closeness criteria as that applied in ToMe at the single frame level currently) for fairness"
>
> We implemented the requested "Temporal ToMe" baseline, which uses cosine similarity for inter-frame reuse. To ensure a comprehensive evaluation, we compared both the specific temporal token reusing modules and the full frameworks. Specifically:
>
> * **ToMe (Temporal):** Reuses tokens across frames based on cosine similarity (as requested).
> * **ToMe (Spatial + Temporal):** Combines standard spatial ToMe with the temporal extension above.
> * **Ours(Temp):** Uses only our Temporal Token Reuse module.
> * **Ours (Full):** Integrates both Temporal Token Reuse and Task-Driven Token Pruning.
>
> As shown in the table below, our method consistently outperforms the ToMe-based equivalents.
>
> * **Temporal Comparison:** **Ours(Temp)** achieves a significantly higher Success Rate than "Temporal ToMe" at a similar inference speed.
> * **Full Framework Comparison:** **Ours (Full)** further demonstrates superior performance compared to the fully combined ToMe baseline.
>
> | Method                    | FPS (Orin) | Success Rate (%) |
> |:------------------------- |:----------:|:----------------:|
> | ToMe (Temporal)           | 46.5       | 43.30            |
> | **Ours (Temp)**           | **47.2**   | **51.20**        |
> | ToMe (Spatial + Temporal) | 48.7       | 40.30            |
> | **Ours (Full)**           | **56.4**   | **50.05**        |

---

> > ### Author Response · Authors · 2025-11-24
> > **Part II**
> >
> > ## W4
> >
> > > "I would also compare performance to a smaller model that is flop matched to the proposed method to check if the pruning method proposed outperforms that at least."
> >
> > As shown in the table below, we trained a smaller ViT-based policy to match the computational cost of our approach.
> > Despite the similar or lower inference speed, the small model achieves a Success Rate of only **35.25%**. This is significantly lower than both the Baseline and our proposed method. This result confirms that dynamically pruning a capable large model retains far better representation quality than training a compact model from scratch.
> >
> > | Method            | FPS (Raspberry Pi) | Success Rate (%) |
> > |:----------------- |:------------------:|:----------------:|
> > | Baseline          | 0.69               | 52.10            |
> > | **Smaller Model** | 1.17               | 35.25            |
> > | Ours (Temp)       | 1.16               | 51.20            |
> > | Ours (Task)       | 1.35               | 51.50            |
> > | **Ours (Full)**   | **1.95**           | **50.05**        |
> >
> > These results have been included in Table 1 of the updated paper.
> >
> > ## W5
> >
> > > "ToMe is a training free approach without much overhead, so the authors should report how much is the overhead from their approach because they have to train two stages on top of the baseline - since if it is comparable compute to training the baseline itself then that is not very meaningful"
> >
> > Training our full framework requires approximately 3 additional days on top of the baseline's 7 days. We argue that this overhead yields significant returns:
> >
> > 1. **Superior Trade-off:** Crucially, this additional training enables us to achieve a **superior efficiency-accuracy trade-off curve** as shown in Figure 3. This performance is **unmatched by existing methods**, covering both training-free approaches like ToMe and training-based ones like DynamicViT. The learned policy supports threshold adjustment during inference without retraining, yielding higher Success Rates at similar inference speeds across the entire curve.
> > 2. **Lifetime Value:** We emphasize the distinction between **one-time offline cost** and **lifetime online benefit**. In robotics, on-device inference latency is the critical constraint. Investing 40% more offline training time to gain a permanent **3x online speedup** on edge devices is a favorable trade-off for practical deployment.
> >
> > ## W6
> >
> > > "I see no change in performance in table 3b for temporal ablations when varying the threshold - maybe we need to ablate the range further to get some insights from this ablation?"
> >
> > We expanded the ablation range for the **Temporal Reuse Threshold** ($thr_{temp}$) as suggested. As shown in the table below, the results demonstrate a clear **trade-off**.
> >
> > As the thresholds decrease, the system becomes more aggressive in reduction, leading to higher FPS but lower Success Rate. This confirms that the threshold parameters effectively control the balance. We clarify that our default value (0.5) was selected as a **representative operating point** to facilitate standardized comparisons with baselines, rather than implying a single optimal setting.
> >
> > | Threshold        | 0.1  | 0.3  | 0.4  | **0.5**  | 0.6  | 0.7  | 0.9  |
> > |:---------------- |:----:|:----:|:----:|:--------:|:----:|:----:|:----:|
> > | Success Rate (%) | 42.6 | 49.0 | 50.0 | **50.1** | 50.6 | 51.1 | 51.8 |
> > | FPS              | 57.5 | 57.1 | 56.7 | **56.4** | 55.1 | 49.3 | 41.6 |
> >
> > These results have been included in Table 3b of the updated paper.
> >
> > ## W7
> >
> > > "Writing needs to be improved to make the paper more self contained and to make the experimental section easier to parse. The experimental section does not define (sp) and (temp) used in the main results table anywhere. There should also be a description of what the different task subsets (base/long) are actually testing and how they differ."
> >
> > We thank the reviewer for pointing this out.
> >
> > We revised the paper to explicitly define `(sp)` as "Task-Driven Pruning only" and `(temp)` as "Temporal Reusing only".
> >
> > Regarding the task subsets, we clarify that our evaluation is performed on the validation set of **HM3D-Semantics v0.1**, following the standard **Habitat Challenge 2022** ObjectNav configuration. We will add these details to the experiment section to ensure clarity.
> >
> > **References**
> >
> > [1] Rao et al. "Dynamicvit: Efficient vision transformers with dynamic token sparsification."  NeurIPS 2021.
> >
> > [2] Bolya et al. "Token Merging: Your ViT But Faster." ICLR 2023.

---

### Author Response · Authors · 2025-12-02
**Summary of Rebuttal**

Dear ACs, SACs, PCs, and Reviewers,

Since there will be no author-reviewer discussion phase, we provide this summary to assist your final decision. We are encouraged by the recognition of our work's value across all four reviewers.

## Consensus on Strengths

Reviewers praised our method as **"simple, intuitive" (Fuvt, kYmX)** and **"novel" (kYmX)**, offering a **"general solution" (qd63)** to an **"important problem" (qd63)** in on-device robotics. They highlighted its **"meaningful real-world value" (srF1)**, noting that it **"performs the best" (Fuvt)** among baselines with **"significant acceleration" (srF1)**. Additionally, reviewers appreciated the **"well thought through" (Fuvt)** analysis and the model's **"interpretability" (kYmX)**.

## Resolution of Key Concerns

We summarize below how we have addressed the primary concerns raised by the reviewers (qd63, Fuvt, srF1, kYmX) with new experiments and evidence.

1. **Generalization to Manipulation Tasks (Raised by qd63, srF1, kYmX)**

   * **Concern:** Reviewers requested evidence of applicability beyond ObjectNav.
   * **Resolution:** We extended our evaluation to the **Meta-World** manipulation benchmark (`reach-v3`). Our method achieves a **2.3x speedup** on Jetson Orin compared to the baseline while maintaining a comparable Success Rate. This confirms the framework generalizes effectively to manipulation settings.

2. **Fairness of Baselines & Comparisons (Raised by qd63, kYmX)**

   * **Concern:** Requests for a "Temporal ToMe" baseline and comparison with smaller models.
   * **Resolution:**
     * We implemented "Temporal ToMe" (using cosine similarity). Our method significantly outperforms it in Success Rate (51.2% vs. 43.3%) at similar speeds, validating our multi-criterion strategy.
     * We compared against a FLOP-matched smaller model. It achieves only 35.25% Success Rate (vs. our ~50%), confirming that pruning a large model is superior to training a small model from scratch.
     * We clarified that our method is orthogonal to architecture-based acceleration and focuses on reducing input redundancy.

3. **Extended Ablation Studies (Raised by qd63, Fuvt, srF1)**
   * **Concern:** Requests for more in-depth analysis on thresholds and design choices.
   * **Resolution:**
     * **Threshold Impact Analysis:** We clarified that the trade-off curve was already in Figure 3. We further expanded the ablation range in the rebuttal, confirming that thresholds serve as effective control knobs and our default settings represent a representative operating point.
     * **Layer Choice Analysis:** We conducted ablations on the insertion layers for reuse/pruning modules. Results confirm that our default configuration offers the optimal balance between exploiting low-level temporal redundancy and high-level semantic relevance.

## Conclusion

We believe we have rigorously addressed all questions. Reviewers **qd63 and srF1** explicitly mentioned the manipulation experiment as their major concern. We resolved this by adding the Meta-World evaluation—a solution that **kYmX** validated as having **"solved"** this issue. **Fuvt** also indicated a willingness to raise the score upon seeing these clarifications. Given the solid performance gains, expanded generalization results, and comprehensive ablations, we hope the AC will consider our paper for acceptance.

---

### Meta-Review · Area_Chair_LBd2 · 2026-01-06

**Summary:**

This paper introduces a task-driven token reduction framework for accelerating Vision Transformers in robotics. By combining Temporal Token Reusing to exploit inter-frame redundancy and Task-Driven Token Pruning to filter spatial regions based on current goals, the method achieves a 1.5×–3× speedup on edge devices like Jetson Orin and Raspberry Pi.

During the rebuttal, the authors addressed several key concerns by providing data on Meta-World manipulation tasks (showing a 2.3× speedup), demonstrating that pruning a large model significantly outperforms training a FLOP-matched smaller model (50% vs. 35% Success Rate), and visualizing how the model adaptively shifts focus based on the task.

Despite these updates and a score increase from Reviewer Fuvt, the paper was ultimately recommended for rejection. The decision was primarily informed by the extra training overhead (3 additional days), which may introduce a potential unfair comparison with training-free methods like ToMe. Furthermore, while the authors emphasized that their approach is orthogonal to architectural modifications, they did not provide sufficient experimental evidence to substantiate how the method performs when combined with other model-level optimizations.

**Reviewer Concerns:**

The rebuttal effectively addressed concerns regarding task generalization and baseline fairness. By providing new experiments on the Meta-World manipulation benchmark, the authors demonstrated that the method generalizes beyond navigation. They also resolved doubts about model capacity by showing that pruning a large ViT significantly outperforms training a FLOP-matched smaller model.

However, several concerns remain outstanding. The 3-day additional training overhead was noted as a potential source of unfair comparison against training-free methods like ToMe, which do not require extra optimization. Additionally, while the authors claimed the method is orthogonal to architectural modifications, they failed to provide experimental evidence showing it successfully stacks with other model-level optimizations. Finally, the reliance on static environments remains a limitation for broader robotic deployment.

**Reviewer Scores:**

NA

---

### Decision · Program_Chairs · 2026-01-26

Reject